# Quantitative Approach to Quality Review of Prenatal Ultrasound Examinations: Estimated Fetal Weight and Fetal Sex

**DOI:** 10.3390/jcm13226895

**Published:** 2024-11-16

**Authors:** C. Andrew Combs, Ryan C. Lee, Sarah Y. Lee, Sushma Amara, Olaide Ashimi Balogun

**Affiliations:** 1Pediatrix Center for Research, Education, Quality & Safety, Sunrise, FL 33323, USA; 2Obstetrix of California, Los Gatos, CA 95032, USA; 3Eastside Maternal-Fetal Medicine, Bellevue, WA 98004, USA; 4Obstetrix Maternal-Fetal Medicine Specialists, Houston, TX 77054, USA; olaide.ashimibalogun@pediatrix.com

**Keywords:** diagnostic error, fetal biometry, fetal growth restriction, large for gestational age, small for gestational age

## Abstract

**Background/Objectives**: Systematic quality review of ultrasound exams is recommended to ensure accurate diagnosis. Our primary objectives were to develop a quantitative method for quality review of estimated fetal weight (EFW) and to assess the accuracy of EFW for an entire practice and for individual personnel. A secondary objective was to evaluate the accuracy of fetal sex determination. **Methods:** This is a retrospective cohort study. Eligible ultrasound exams included singleton pregnancies with live birth and known birth weight (BW). A published method was used to predict BW from EFW for exams with ultrasound-to-delivery intervals of up to 12 weeks. Mean error and median absolute error (AE) were compared between different personnel. Image audits were performed for exams with AE > 30% and exams with reported fetal sex different than newborn sex. **Results:** We analyzed 1938 exams from 890 patients. In the last exam before birth, the median AE was 5.9%, and the predicted BW was within ±20% of the actual BW in 97.2% of patients. AE was >30% in 28 exams (1.4%); image audit found correct caliper placement in all 28. Only two patients (0.2%) had AE > 30% on the last exam before birth. One sonographer systematically over-measured head and abdominal circumferences, leading to EFWs that were overestimated. Reported fetal sex differed from newborn sex in seven exams (0.4%) and five patients (0.6%). Images in four of these patients were annotated with the correct fetal sex, but a clerical error was made in the report. In one patient, an unclear image was labeled “probably female”, but the newborn was male. **Conclusions:** The accuracy of EFW in this practice was similar to literature reports. The quantitative analysis identified a sonographer with outlier measurements. Time-consuming image audits could be focused on a small number of exams with large errors. We suggest some enhancements to ultrasound reporting software that may help to reduce clerical errors. We provide tools to help other practices perform similar quality reviews.

## 1. Introduction

Estimation of fetal weight is an essential component of the standard obstetrical examination in the second and third trimesters of pregnancy [1]. Estimated fetal weight (EFW) derived from ultrasound biometry is central to the diagnosis of abnormal fetal growth, both fetal growth restriction (FGR) and large-for-dates. However, sonographic EFW is known to be imperfect; over 20% of EFWs differ from BW by more than 10% [2]. Such a high rate of relatively large errors results in a high potential for misdiagnosis of abnormal fetal growth.

Misdiagnosis of abnormal fetal growth has clinical consequences. For pregnancies with FGR, fetal surveillance is recommended, including serial assessments of fetal well-being, umbilical artery Doppler ultrasound, and fetal growth; with severe FGR, early delivery is recommended [3]. These interventions all have both direct financial costs and indirect costs of patient time, inconvenience, and loss of work. For pregnancies suspected to be large-for-dates, induction of labor is sometimes recommended, and the rate of cesarean delivery is increased, even if actual birth weight (BW) is not increased [4,5,6,7,8,9,10,11,12].

Various professional organizations recommend systematic quality reviews to ensure the accuracy of obstetrical ultrasound diagnoses [13,14,15,16]. We recently presented a method for a quantitative review of fetal biometric measurements, comparing the z-scores of measurements obtained by individual sonographers within a practice [17]. We demonstrated how image audits could be focused on sonographers whose mean z-scores deviated from the practice-wide mean, thereby saving the time and expense of auditing those whose measurements fell within the expected range. If sonographers vary in the accuracy of their biometric measurements, it seems likely that they might also vary in the accuracy of their EFWs.

The primary aim of the present study was to develop and demonstrate methods to assess the accuracy of EFW for an ultrasound practice overall and for individual providers. A secondary aim was to assess the accuracy of fetal sex determination in prenatal ultrasound exams.

## 2. Materials and Methods

### 2.1. Study Design and Setting

A retrospective quality review was undertaken for the cohort of ultrasound exams performed from January 2022 to December 2023 at Obstetrix of California, a suburban, referral-based maternal–fetal medicine (MFM) practice. During the study period, the practice used GE Voluson ultrasound machines (GE Healthcare, Wauwatosa, WI, USA) with variable-frequency curvilinear transducers (2 to 9 mHz as needed to optimize imaging). We used GE Viewpoint software (Version 6) to generate reports and store images and metadata. Before starting this study, the protocol was reviewed by the Institutional Review Board (IRB) of Good Samaritan Hospital (GSH, San Jose, CA, USA). A retrospective study of existing data was determined to be of minimal risk to patients and was granted Exempt status (#FWA-00006889, 11 October 2023).

### 2.2. Data Extraction and Inclusion and Exclusion Criteria

Using the Viewpoint query tool, we extracted data for all ultrasound exams meeting these eligibility criteria: fetal cardiac activity present and 4 basic fetal measurements recorded: biparietal diameter (BPD), head circumference (HC), abdominal circumference (AC), and femur length (FL). The extracted data for each exam included patient identifiers (name, date of birth, person number), exam identifiers (date, exam number), sonographer who performed the exam, physician who interpreted it, referring provider name, exam date, best obstetrical estimate of due date (EDD) [17], fetal measurements, and fetal sex.

We divided patients into 3 groups based on the referring obstetrical provider: (1) those with likely delivery at GSH because the referring physician had delivery privileges there; (2) those with possible delivery at GSH because the provider was from a distant hospital without neonatal intensive care, possibly reflecting a maternal transport; and (3) those unlikely to deliver at GSH because the provider practiced primarily at another hospital. From the first 2 groups, we created a spreadsheet containing patient identifiers and EDD but no other exam data. An investigator reviewed the GSH electronic record system to locate the patient and to record the delivery date, birth weight (BW), newborn sex, mode of delivery, admission to neonatal intensive care (yes/no), any congenital anomalies identified at birth, and the self-reported maternal race. This review was performed without access to any of the ultrasound findings. The file with delivery data was then merged with the data file containing the ultrasound exam information.

For analysis, we excluded multifetal pregnancies, stillbirths, and cases with gastroschisis or omphalocele. We excluded cases with extreme outliers for HC, AC, or FL (values more than 6 standard deviations (SD) from the normal mean based on formulas derived from World Health Organization norms) [18,19,20,21]. We also excluded deliveries with an implausible combination of EDD and delivery date, i.e., a calculated gestational age at birth < 0 or >50 weeks; most of these were attributed to a repeat pregnancy for which we did not have ultrasound data.

For patients with more than 1 eligible exam during the study period, all exams were included. Separate analyses were restricted to only the last exam, i.e., the shortest interval from ultrasound to delivery (latency).

### 2.3. Calculation of EFW, Errors, and Predicted BW

EFW was calculated from HC, AC, and FL using the 3-parameter formula of Hadlock et al. [22] because 2 large comparative reviews concluded that this is the most accurate of over 70 formulas for calculating EFW [2,23].

The initial analysis was focused on exams performed within 1 day before birth, where the accuracy of EFW was assessed by comparing it to birth weight (BW). Taking EFW as the predicted BW (BW_predicted_), with error expressed as a percentage:Error = 100 · (BW_predicted_ − BW)/BW
Absolute Error = |Error|
where · signifies multiplication. Positive values of error reflect overestimation of EFW, and negative values reflect underestimation. Absolute error reflects only the magnitude of the error, ignoring its direction.

Because the number of ultrasound exams within 1 day of birth was too small for meaningful comparisons between sonographers, we used the method of Mongelli and Gardosi [24] to predict BW from EFW in exams with latencies of several weeks. The method predicts BW based on the assumption that the ratio of EFW to median normal fetal weight at the gestational age of the ultrasound exam (GA_us_) stays constant over time and should therefore equal the ratio of BW to the median normal fetal weight at the gestational age of birth (GA_birth_). This can be simplified to the formula:BW_predicted_ = EFW · e^x^
where x = −0.00354 · (GA_birth_^2^ − GA_us_^2^) + 0.332 · (GA_birth_ − GA_us_)
where GA_us_ and GA_birth_ are expressed in exact weeks and e is the Euler constant (approximately 2.718). The derivation and validation of this formula are given in Appendix A. We also show in Appendix A that the assumption of a constant weight ratio is equivalent to assuming that the EFW percentile and z-score remain constant from ultrasound to birth. As shown in Appendix A, the z-score of EFW is highly correlated with the z-score of BW for latencies < 12 weeks (r = 0.82, 0.71, 0.66 at latency of 0–3.9 weeks, 4–7.9 weeks, and 8–11.9 weeks, respectively). Because of decreasing accuracy with increasing latency, exams with latencies ≥ 12 weeks were excluded from the analysis.

### 2.4. Statistical Analyses

Analyses were performed with Stata (version 18, Statacorp, College Station, TX, USA). Two-tailed P-values less than 0.05 were considered significant.

Errors were normally distributed and are presented as mean ± standard deviation (SD). Between-group errors were compared using one-way analysis of variance (ANOVA) with the Sidak multiple-comparisons test. Absolute errors were not normally distributed and are presented as median with interquartile range (IQR). Between-group comparisons of absolute error were performed with Kruskal–Wallis and U-tests. We compared errors and absolute errors between exams of differing latency (in 4-week blocks) and between individual sonographers and physicians.

Several exploratory analyses were performed to evaluate possible factors that might influence the accuracy of BW_predicted_ including maternal race, maternal obesity, maternal age, newborn sex, gestational age at ultrasound or birth, birth weight, or fetal weight percentile. These are presented in Appendix A.

### 2.5. Audit of Exams with Large Errors

The protocol reviewed by the IRB specified that audits would be performed for exams with an absolute error of more than 40%. We expanded the audits to all exams with an absolute error of more than 30% because the number of such exams was small. Each audit started with a manual double-check of the hospital record to confirm that the BW and delivery date were entered correctly in our data file. Then, we reviewed ultrasound reports and images to assess whether there were measurement inaccuracies, errors in transcription, or other factors to explain the error.

Similar audits were performed for all cases in which fetal sex on the ultrasound exam report was different than the newborn sex recorded in the delivery records and for the one case excluded because of an extreme outlier biometric value.

## 3. Results

### 3.1. Summary of Included and Excluded Exams

A total of 1938 ultrasound examinations in 890 patients met all the inclusion criteria and were included in the analyses. The flow diagram in Figure 1 shows the progress of exams from initial eligibility to final inclusion.

### 3.2. Accuracy of EFW for Exams Within 1 Day of Birth

Ultrasound was performed within one day of birth in 53 patients. For these, the birthweight averaged 2819 ± 782 gm (mean ± SD) at gestational age 36.2 ± 3.4 weeks. The difference between EFW and BW averaged −88 ± 266 gms (mean ± SD) for a mean error of −2.3 ± 8.8% of BW. The negative value of error signifies that EFW tended to be smaller than BW, but the mean error was not significantly different than 0 (*p* = 0.07, *t*-test). The median absolute error was 6.3% (IQR 3.1–9.0%). Absolute errors of less than 10% were seen in 41 cases (77%), less than 20% in 50 cases (94%), and 20–30% in 3 cases (6%). These 53 exams were performed by 17 sonographers who performed from one to five exams each, so meaningful comparisons between sonographers were not feasible. No sonographer had more than one exam with an absolute error > 20%.

### 3.3. Accuracy of Predicted Birthweight for Exams Within 12 Weeks of Birth

The accuracy of BW_predicted_ at different latencies is summarized in Table 1. In every 4-week latency block, BW_predicted_ was higher than BW by about 3-5%; this overestimation was significant in every block and for the overall sample (*p* < 0.001, *t*-test). In every latency block, the median absolute error was ≤7%. Overall, over 71% of predictions were within 10% of BW and over 22% of predictions were within 10–20% of BW, so 93.9% of predictions were within 20% of BW. Even with a latency of 8–11.9 wks, 89.8% of predictions were within 20% of BW.

As shown in the lower half of Table 1, when the analysis was restricted to the last ultrasound exam before birth, there was a lower mean error and a lower median percent error, and 97.2% of the predictions were within 20% of BW. Moreover, there was no significant difference in accuracy between different latency blocks in this subset, that is, exams with a latency of 8–11.9 weeks had similar accuracy to those with a latency of 0–3.8 weeks.

### 3.4. Accuracy of Predicted Birth Weight for Individual Sonographers

The accuracy of predicted BW for seven individual sonographers is summarized in Table 2. To minimize the possibility of spurious values with a small sample size, we restricted the Table to sonographers with more than 100 EFWs during the measurement period. All these sonographers had mean errors significantly above 0 (BW_predicted_ higher than BW). Sonographer 9 had a significantly larger mean error and mean absolute error than the other sonographers.

The lower half of Table 2 summarizes the accuracy of the last exam before birth. The mean error and median absolute error were smaller in this subset than in the All Exams cohort, and 97.2% of predictions were within 20% of BW. Although Sonographer 9 still had a higher mean error and absolute error than the others, the difference did not reach significance in this subset.

We considered two possible reasons why Sonographer 9 had a larger mean error than the others: first, if the exams had a longer latency interval between ultrasound and birth than other sonographers’ exams, a larger error might be expected; second, if fetal biometry was systematically over-measured, the EFWs would have larger error. The mean latency and biometry z-scores for the seven sonographers with at least 100 exams each are summarized in Table 3. The mean latency for Sonographer 9 was slightly higher than for the others, but not significantly so. Using the z-score method for the assessment of biometry that we previously described [25], Sonographers 9 and 24 systematically over-measured HC and AC (z-scores significantly larger than the practice mean). Sonographers 16 and 22 also systematically over-measured HC. Sonographers 9 and 24 are no longer with the practice, so no further audit was performed for them.

### 3.5. Accuracy of Predicted BW for Individual Physicians

In most cases, biometry measurements are obtained by sonographers. If multiple measurements of the same biometric parameter are obtained, the ultrasound machine will automatically report the average measurement to Viewpoint software. Physicians may sometimes repeat the measurements or may select a particular measurement for reporting, overriding the automated average. If a physician does this repeatedly, it is possible that their EFWs will be more accurate or less accurate.

Table 4 evaluates the possibility that physicians might differ in the accuracy of their BW predictions. Physician 6 had a slightly lower mean error than the others but no significant difference in the median absolute error. The difference in the mean error was still evident but no longer significant in the last exam before birth. Physician 6 is known to change the sonographers’ measurements more often than the other physicians, and this appears to have been associated with a slight improvement in accuracy.

### 3.6. Exploration of Other Factors Potentially Affecting the Accuracy of Predicted BW

Appendix A contains analyses of the accuracy of BW predictions stratified by maternal race, maternal obesity, maternal age, newborn sex, gestational age, and fetal weight. Some of these factors had statistically significant associations with accuracy (e.g., the mean error was slightly lower in Hispanic patients than in Asian or White patients, lower in male than female fetuses, and higher with ultrasound exams before 30 weeks of gestation; the absolute error was higher among preterm births), but none of these was large enough to have a major bearing on the quality review. We found no association between accuracy and maternal obesity or maternal age.

### 3.7. Diagnostic Accuracy of Prediction of Small- or Large-for-Gestational Age (SGA or LGA)

Appendix A tabulates the test performance characteristics of using FGR (defined as either EFW or AC < 10th percentile to predict SGA (BW < 10th percentile) or using EFW > 90th percentile to predict LGA (BW > 90th percentile)). These analyses were based on the last exam before birth.

Diagnosis of FGR had low sensitivity (51%) but high specificity (92%) for the prediction of SGA. When the analysis was restricted to exams < 7 days before birth, sensitivity was improved (84%) but specificity was lower (83%). The areas under the receiver operating characteristic (AUROC) curves relating the EFW percentile to SGA were 0.88 and 0.93, respectively.

EFW > 90th percentile had low sensitivity and high specificity for the prediction of LGA, irrespective of whether the analysis included all exams (43%, 97%, respectively) or only exams < 7 days before birth (44%, 95%). The AUROCs were 0.91 and 0.95, respectively.

### 3.8. Audit of Exams with Large Errors

We audited the single exam that was excluded because of an extreme outlier biometric value (FL more than 14 SD below the mean). On this exam, the FL was recorded as 3.2 mm, but the images showed an FL of 3.21 cm (i.e., a 10-fold error).

We audited all exams with an absolute error of more than 30% (28 exams in 17 patients). Image audit found acceptable caliper placement for biometric measurements in all 28 of these exams. In 15 patients, subsequent exams showed absolute errors < 30%, so absolute errors > 30% were seen on the last exam before birth in only 2 of 890 patients (0.2%). For 14 of the other 15 patients, BW_predicted_ was higher than BW (errors > 30%) on an early exam but subsequent exams showed progressively smaller errors; fetal growth restriction was ultimately diagnosed in 10 of these patients. For the 15th patient, BW_predicted_ was lower than BW (error < −30%); this was an ultrasound exam at 26 weeks of gestation in a patient with Type 1 diabetes; BW_predicted_ was 3021 gm, but BW was 4620 gm at 37 weeks (latency 11 weeks).

### 3.9. Accuracy of Fetal Sex Reporting

Differences between newborn sex and fetal sex recorded with the ultrasound exam were found in 7 of 1938 exams (0.4%) and 5 of 890 patients (0.6%), as summarized in Table 5. The audit found that the correct fetal sex was seen and labeled on the images for Cases 1 through 4. For one patient (Case 2), the error reported on the first exam was carried forward to two later exams even though the genital area was not imaged on the follow-up exams. In one exam (Case 5), the image was unclear but labeled as “probably female” and recorded as female in the database; fetal sex was not disclosed to the patient or included in the report to the referring provider, at the patient’s request; the newborn was male.

In addition, fetal sex was recorded as “uncertain” in 13 exams (0.7%) from 11 patients (1.2%). These were performed by seven different sonographers, with each having only 1–3 exams with uncertain fetal sex, a number too small to suggest a systematic problem. For four of these exams (three patients), the correct sex was recorded in another exam of the same patient. In four exams (three patients), the fetus was in breech presentation with GA_us_ > 30 weeks, so fetal genitalia could not be seen. For the remaining five exams (five patients), no specific reason was given for the uncertainty; three of these had other exams (not included in the analysis because of long latency) where sex was also recorded as uncertain. None of the newborns had ambiguous genitalia.

## 4. Discussion

### 4.1. Principal Findings

In this practice, EFW obtained up to 12 weeks before birth yielded accurate predictions of BW. In the last exam before birth, the median absolute error was less than 6%, over 75% of predictions were within 10% of BW, and over 97% were within 20% of BW. These results are comparable to those reported in a study of 5163 exams performed within 2 days of birth, where absolute error averaged 6.7%, and 77.7% of EFWs were within 10% of BW [2].

Between-sonographer comparisons revealed one sonographer whose mean EFW was significantly overestimated compared to the others, attributable to the overmeasurement of HC and AC. Between-physician comparisons revealed one physician whose mean EFW was slightly less overestimated than the others, but there was no significant difference in the median absolute error. We are not aware of any prior studies comparing the accuracy of individual sonographers or physicians.

The audit of exams with large errors did not reveal any images with incorrect caliper placement. Only 2 patients of 890 (0.2%) had ≥30% error on the last exam before birth. In one excluded case with an extreme outlier measurement, the FL was recorded in centimeters rather than millimeters, a clerical error.

Fetal sex was incorrectly recorded in 0.4% of exams (0.6% of patients). In one case, the sonographer was uncertain about fetal sex and recorded an incorrect guess. In all other cases, the sonographer had correctly determined the fetal sex, but a clerical error was made in documenting the findings.

### 4.2. Prediction of Birthweight After Long Latency

The assumption that the fetal weight percentile remains constant as the fetus grows yielded very accurate predictions of birth weight in most cases. Most of the exceptions (i.e., those with absolute errors ≥ 30%) were cases with progressive fetal growth restriction (n = 14) or evolving fetal macrosomia with diabetes (n = 1). When the analysis was restricted to the last exam before birth, only two patients (0.2%) had an error ≥ 30%.

Despite low mean errors and median absolute errors, however, the diagnostic accuracy of abnormal fetal size (FGR or EFW > 90th percentile) had low sensitivity for predicting abnormal BW (SGA or LGA).

Diagnostic performance for predicting SGA in our results was similar to the performance recently reported in a series of 10,045 patients (sensitivity 51% vs. 36%, specificity 92% vs. 96%, respectively) [26]. In our patients, sensitivity was substantially higher in exams < 7 days before birth (84%), though with decreased specificity (83%).

Diagnostic performance for predicting LGA in our results was similar to the performance recently reported in a series of 26,627 exams (sensitivity 43% vs. 41%, specificity 97% vs. 95%, respectively) [27].

Ultimately, the reason for diagnosing abnormal growth is not to predict whether BW will be abnormal but whether the fetus will be at risk for morbidity [28]. To improve sensitivity for the prediction of risk of perinatal morbidity, different cut-off values than the traditional 10th and 90th percentiles may be needed, but different cut-offs may also result in decreased specificity [28].

We have shown that the prediction of BW after long latency is useful for quality review in that it allows for sample sizes sufficient to detect subtle inaccuracies in EFW determination. However, we do not encourage clinical diagnoses such as FGR or EFW > 90th percentile to be made based on long-ago exams. Accuracy and diagnostic performance degrade with increasing latency, so clinical management decisions should be based only on an up-to-date EFW.

### 4.3. Quantitative Analysis to Guide Focused Image Audit

The traditional approach to quality review of ultrasound exams involves auditing images for errors in measurement and interpretation. For example, in the RADPEER program of the American College of Radiology (ACR), a percentage of examinations are randomly selected for “double-reading” by a second physician [29,30]. This approach is time-consuming and rarely uncovers clinically significant errors [30,31,32,33]. The RADPEER program is targeted primarily on structural findings rather than measurements. For fetal biometry, a similar method for quality monitoring was used by the INTERGROWTH-21st project in its international prospective study of fetal growth [34]. In that program, a 10% random sample of images was sent to a central Quality Unit for a second reading. This comprehensive method is laudable but is likely too time-consuming and labor-intensive for a typical clinical practice.

In contrast, using our quantitative approach, the accuracy of EFW can be assessed without any reference to images at all. One needs to only compare the actual BW to the BW predicted from EFW. While EFW, in turn, is derived from biometric measurements on the images, there is little value in auditing biometry images from the majority of exams in which EFW is reasonably accurate. Image audits can be targeted at a small number of exams with large errors.

We demonstrate how sonographers (Table 2) or physicians (Table 4) whose accuracy deviates from other personnel in the practice can be identified using the quantitative approach and standard statistical methods. For the one sonographer whose EFWs were systematically over-measured, we demonstrate how a quantitative assessment of BPD, HC, AC, and FL can reveal systematic measurement deviations (Table 3) using methods we have described previously [25]. If the deviations are large enough, image audits can be targeted to those specific views.

The focused audits revealed several clerical errors. In a single case, the FL was recorded in centimeters rather than millimeters; this error must have been made by someone entering the FL manually because the ultrasound machines transmit biometry measurements to Viewpoint software automatically. Similarly, manual entry of fetal sex into the database was the likely cause in most of the cases of discordance between reported fetal sex and birth sex. Human error is a known contributor to misdiagnosis [35] and is a potential hazard for any finding that must be entered manually. To reduce the need for manual re-entry of known data, Viewpoint software automatically carries some findings forward to the next exam. However, we found one patient where the incorrectly assigned sex was carried forward automatically to subsequent exams even though fetal genitalia were not imaged again, compounding the error rather than reducing it.

### 4.4. Measurement Quality Review in Context

A review of the accuracy of EFW is only one component of a comprehensive quality program for prenatal ultrasound. Other components include a review of fetal biometry, which we have previously described [25], and a review of fetal anatomy imaging. In a forthcoming article, we will describe our quantitative approach to quality review for the fetal anatomy survey.

There are also several structural components generally recommended for a high-quality ultrasound practice [13,14,15,16]. These include accreditation of the practice by an organization such as the American Institute of Ultrasound in Medicine [15] or ACR [16]. Accreditation requires that all personnel have formal initial education and training, continuing education, certification, or licensing. Accreditation also requires written protocols to ensure uniformity of exams, timely communication of results, cleaning and disinfection of transducers, equipment maintenance, and patient safety and confidentiality. An onboarding program is recommended for new personnel, including orientation to practice protocols and assessment of competency [13,14].

### 4.5. Clinical Implications

Prior to performing this systematic quality review, we had no way of knowing whether our EFWs were accurate or whether accuracy varied by sonographer or by physician. We would occasionally hear from referring providers about BWs that were radically different than expected, but such reports are obviously a biased sample.

Because EFWs obtained up to 12 weeks before birth had accuracy and diagnostic performance similar to published findings from studies with much shorter latencies [2,26,27], we can report EFWs to patients and referring providers with reasonable confidence, with a median absolute error of 6% or less (Table 1, lower half). Nonetheless, we must still exercise caution in basing clinical decisions on EFWs because up to one-fourth of EFWs can be in error by 10% or more.

Had we identified actionable differences in accuracy between personnel, this would have formed the basis for a quality improvement process that would have started with a one-on-one review with individual personnel to educate them on the importance of accurate EFW determination and to review measurement techniques, as outlined in our article on quality review of fetal biometry [25]. The goals of such a quality improvement process would be to improve accuracy and to increase standardization across the practice.

### 4.6. Strengths and Limitations

A strength of the quantitative summary is that it provides a large-scale overview of an entire practice and the accuracy of individual personnel (Table 1, Table 2, Table 3, Table 4 and Table 5). Analyzing a large number of exams for each provider, the method is highly sensitive to small between-provider variations, readily identifying outlier personnel for focused review. Once the analysis script is written, the quantitative analysis can be performed rapidly and repeated periodically. For those who wish to use these methods, we have provided a sample Excel spreadsheet with pseudodata (Appendix A) and sample Stata scripts that will perform the basic analyses for Table 1, Table 2, Table 3, Table 4 and Table 5 (Appendix A).

An important limitation was that we needed to retrieve BW and other delivery data by manual review of the maternal hospital record, a time-consuming endeavor. We are not part of an integrated health system with an electronic health record system that automatically links maternal and newborn information. Only 20% of our potentially eligible patients were from referring providers who deliver routinely at our primary hospital, and we were only able to find delivery information on 75% of those (Figure 1). When we were unable to match patients by full name and date of birth, we were sometimes successful in matching first name plus date of birth because first names generally do not change during the course of a pregnancy but surnames frequently change. Some patients probably delivered elsewhere, but we doubt that this would account for our failure to match 25% of patients. All told, it took about 50 person-hours to perform the hospital record search and review. Less time would certainly have been required if our record system was integrated with the hospital’s or if an automated database retrieval program had been available.

Another limitation is that some sonographers had a relatively small number of exams, even though the review period was 2 full years. We reviewed their results but did not include them in the Tables because small N’s can produce spurious results. Increasing the study duration to 3 or 4 years might have allowed for a sufficient sample for some of those sonographers. On the other hand, a large number of exams can result in very low *p*-values even with relatively small errors. For example, Sonographer 8 had a mean error of 2.0% in the last exam before birth (Table 2, lower half), which was significantly different than 0 (BW_predicted_ > BW, *p* = 0.017); in a typical 3500 gm newborn, an error of 2% is 70 gm, too small to be of clinical importance.

Another limitation is the inherent time lag between the ultrasound exam and the quality review, including both the latency from ultrasound to birth and the time it takes to accumulate sufficient cases for review. In the interim, there can be personnel turnover, as evidenced by the sonographers with outlier measurements who had left the practice before their systematic overmeasurement was detected. This limitation would naturally be exacerbated by increasing the number of years under study.

### 4.7. Future Directions—Software Enhancements

Viewpoint ultrasound reporting software includes data fields to record birth weight and delivery date. Other software may have similar fields. We suggest that software developers provide tools to link these newborn fields to prenatal ultrasound exams. Tools should also be developed to automate quality review of the accuracy of birth weight predictions from EFW for individual personnel and for the whole practice, including the mean error, median absolute error, and tabulation of exams with various degrees of error, similar to our Table 1, Table 2 and Table 4. Although we have provided a sample data file and analysis script in Appendix A to guide those who wish to replicate our methods, software developers and vendors are more likely than individual medical practices to have the resources to implement and standardize these tools.

Minor software enhancements might help to prevent some clerical errors. First, findings from a prior exam should not automatically carry forward to subsequent exams. We found one patient for whom the same erroneous fetal sex was carried forward to two subsequent exams despite lack of any imaging of the fetal genitalia. On the other hand, useful information that can be carried forward from one exam to the next might include a reminder to re-evaluate views that were suboptimal on the earlier exam, such as uncertainty about fetal sex or other elements of the anatomy, perhaps displaying incomplete items in red. Second, although it is important to allow the manual override of automatically transferred biometrics, the reporting software should flag extreme outliers as a crosscheck, perhaps by displaying values in red. We had one case where an erroneous measurement was off by a factor of 10 (presumably manually entered), resulting in an EFW < 1st percentile; this was not noticed, and the report indicated “fetal size appropriate for dates”.

## 5. Conclusions

We conclude that EFWs performed in this practice with latencies up to 12 weeks have accuracy similar to those reported in the literature for exams with latencies of only a few days. We identified one sonographer who systematically over-measured HC and AC, and whose EFWs therefore yielded BW_predicted_ that was, on average, higher than actual BW. In a practice-wide audit of the few exams with large errors, caliper placement was uniformly acceptable. Ultrasound diagnosis of fetal sex was correct in 99.9% of patients, but several clerical errors resulted in reports that were correct in only 99.4%.

We believe it is important for every ultrasound practice to perform a quality review such as this. We have demonstrated a method and provided tools to allow practices to perform a similar evaluation.

## Figures and Tables

**Figure 1 jcm-13-06895-f001:**
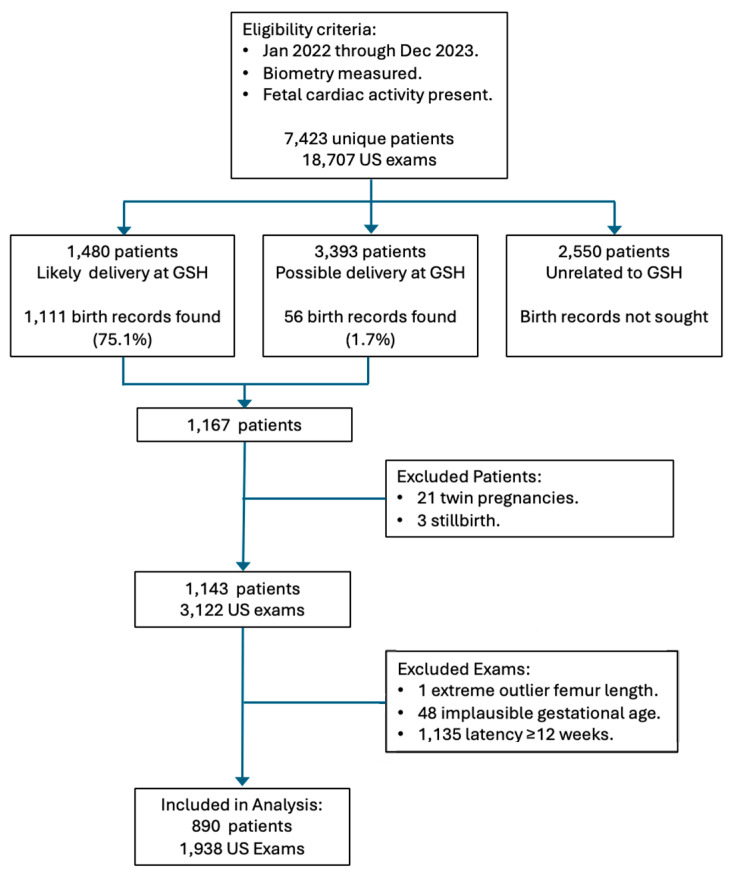
Flow diagram showing numbers of patients, eligible exams, exclusions, and exams included in the final analysis. Abbreviations: GSH—Good Samaritan Hospital. US—ultrasound.

**Table 1 jcm-13-06895-t001:** Accuracy of birth weight predictions at different latencies for the entire practice.

Latency	N	Percent Error, Mean ± SD	Percent Absolute Error, Median (IQR)	Exams with Absolute Error Less than 10%, n (%)	Exams with Absolute Error 10 to <20%, n (%)	Exams with Absolute Error 20 to <30%, n (%)	Exams with Absolute Error 30% or More, n (%)
All Exams							
0–3.9 wks	800	2.9 ± 8.7 ^a^	5.9 (3.1–9.8)	609 (76.1%)	165 (20.6%)	21 (2.6%)	5 (0.6%)
4–7.9 wks	706	3.8 ± 10.0 ^a^	6.4 (3.0–11.3)	490 (69.4%)	169 (23.9%)	37 (5.2%)	10 (1.4%)
8–11.9 wks	432	4.8 ± 11.4 ^ab^	7.0 (3.3–11.9) ^c^	288 (66.7%)	100 (23.2%)	31 (7.2%)	13 (3.0%)
Total	1938	3.7 ± 9.9 ^a^	6.4 (3.1–11.0)	1386 (71.5%)	435 (22.5%)	89 (4.6%)	28 (1.4%)
Last Exam Before Birth							
0–3.9 wks	691	2.8 ± 8.5 ^a^	5.9 (3.1–9.8)	525 (76.0%)	148 (21.4%)	16 (2.3%)	2 (0.3%)
4–7.9 wks	155	2.5 ± 8.7 ^a^	5.7 (2.7–10.2)	116 (74.8%)	35 (22.6%)	4 (2.65)	0
8–11.9 wks	44	2.7 ± 9.8	5.9 (2.8–9.4)	33 (75.0%)	8 (18.2%)	3 (6.8%)	0
Total	890	2.8 ± 8.6 ^a^	5.9 (3.0–9.9)	674 (75.7%)	191 (21.5%)	23 (2.5%)	2 (0.2%)

Latency is the interval from ultrasound exam to birth. ^a^ significantly different than 0, *t*-test, *p* < 0.001. ^b^ significantly different than the 0–3.9 week group, ANOVA with Sidak test, *p* < 0.005. ^c^ significantly different than the 0–3.9 week group, u-test, *p* < 0.005.

**Table 2 jcm-13-06895-t002:** Accuracy of birth weight predictions for sonographers with at least 100 exams.

Sonographer Number	N	Percent Error, Mean ± SD	Percent Absolute Error, Median (IQR)	Exams with Absolute Error Less than 10%, n (%)	Exams with Absolute Error 10 to <20%, n (%)	Exams with Absolute Error 20 to <30%, n (%)	Exams with Absolute Error 30% or More, n (%)
All Exams							
6	212	5.3 ± 8.6 ^a^	6.8 (3.3–11.4)	147 (70.0%)	51 (24.3%)	11 (5.2%)	1 (0.5%)
8	225	2.5 ± 9.5 ^a^	6.3 (2.9–11.2)	157 (70.1%)	58 (25.9%)	7 (3.1%)	2 (0.9%)
9	127	6.8 ± 11.2 ^ab^	8.3 (3.6–14.5) ^c^	81 (62.8%)	34 (26.4%)	9 (7.0%)	5 (3.9%)
16	154	3.1 ± 9.4 ^a^	6.2 (3.3–9.5)	118 (77.6%)	27 (17.8%)	5 (3.3%)	2 (1.3%)
17	145	2.7 ± 9.4 ^a^	5.3 (2.8–9.3)	115 (79.3%)	23 (15.9%)	2 (1.4%)	5 (3.5%)
22	267	3.5 ± 10.8 ^a^	6.3 (2.6–11.2)	190 (71.2%)	61(22.9%)	11 (4.1%)	5 (1.9%)
24	225	5.4 ± 9.5 ^a^	6.7 (3.2–11.3)	158 (70.2%)	49 (21.8%)	17 (7.6%)	1 (0.4%)
Total	1938	3.7 ± 9.9 ^a^	6.4 (3.1–11.0)	1386 (71.5%)	435 (22.5%)	89 (4.6%)	28 (1.4%)
Last Exam Before Birth							
6	90	5.0 ± 7.6 ^a^	6.4 (3.3–9.8)	68 (76%)	20 (22%)	2 (2%)	0
8	105	2.0 ± 8.6 ^a^	6.0 (3.3–10.2)	77 (73%)	27 (26%)	0	1 (1%)
9	44	6.0 ± 8.9 ^ab^	7.6 (3.5–11.8) ^c^	31 (70%)	11 (25%)	2 (5%)	0
16	76	2.1 ± 7.3 ^a^	5.8 (3.3–8.4)	61 (80%)	15 (20%)	0	0
17	58	1.4 ± 7.9	5.0 (3.3–9.1)	47 (91%)	10 (17%)	1 (2%)	0
22	105	2.3 ± 8.3 ^a^	5.5 (2.6–9.8)	81 (77%)	22 (21%)	2 (2%)	0
24	84	5.2 ± 7.9 ^a^	5.9 (3.1–9.4)	67 (80%)	12 (14%)	5 (6%)	0
Total	890	2.8 ± 8.6 ^a^	5.9 (3.0–9.9)	674 (75.7%)	191 (21.5%)	23 (2.5%)	2 (0.2%)

Total includes 17 additional sonographers with ≤100 exams each. ^a^—significantly different than 0, *t*-test, *p* < 0.05. ^b^—significantly different than Sonographer 8, Sidak test, *p* = 0.005. ^c^—significantly different than Sonographers 16 and 17, Kruskal–Wallace test, *p* < 0.01.

**Table 3 jcm-13-06895-t003:** Latency and biometry for sonographers with at least 100 exams.

Sonographer Number	N	Latency, Weeks Mean ± SD	Head Circumference z-Score, Mean ± SD	Abdominal Circumference z-Score, Mean ± SD	Femur Length z-Score, Mean ± SD
6	212	5.1 ± 3.0	0.09 ± 1.00	0.19 ± 0.82	0.33 ± 1.07
8	225	5.6 ± 3.1	–0.10 ± 0.88	0.22 ± 0.80	0.16 ± 0.99
9	127	6.0 ± 3.3 ^a^	0.18 ± 0.85 ^b^	0.38 ± 0.74 ^b^	0.25 ± 1.00
16	154	4.9 ± 3.2	0.21 ± 1.04 ^b^	0.20 ± 0.79	0.09 ± 1.00
17	145	5.6 ± 3.1	0.02 ± 0.92	0.18 ± 0.72	0.09 ± 0.67
22	267	5.6 ± 3.2	–0.22 ± 1.03 ^b^	0.15 ± 0.78	0.45 ± 1.10 ^bd^
24	225	5.7 ± 3.2	0.13 ± 0.92 ^b^	0.47 ± 0.85 ^bc^	0.27 ± 0.86
Practice Total	1938	5.2 ± 3.2	–0.01 ± 0.98	0.21 ± 0.82	0.17 ± 1.01

Practice Total includes 17 additional sonographers with ≤100 exams each. ^a^ *p* < 0.05 for overall between-sonographer difference (ANOVA), *p* = 0.08 comparing Sonographers 9 vs. 16 (Sidak test). ^b^ *p* < 0.05 compared to practice mean (*t*-test). ^c^ *p* < 0.05 compared to Sonographers 6, 8, 15, 17, and 22 (ANOVA with Sidak test). ^d^ *p* < 0.05 compared to Sonographers 8, 16, and 17 (ANOVA with Sidak test).

**Table 4 jcm-13-06895-t004:** Accuracy of birth weight predictions by interpreting physicians.

Physician Number	N	Percent Error, Mean ± SD	Percent Absolute Error, Median (IQR)	Exams with Absolute Error Less than 10%, n (%)	Exams with Absolute Error 10 to <20%, n (%)	Exams with Absolute Error 20 to <30%, n (%)	Exams with Absolute Error 30% or More, n (%)
All Exams							
1	164	4.8 ± 10.6 ^a^	7.0 (3.0–11.5)	113 (68.9)	38 (23.2)	9 (5.5)	4 (2.4)
2	88	5.2 ± 9.1 ^a^	6.8 (2.9–12.6)	61 (69.3)	24 (27.3)	2 (2.3)	1 (1.1)
3	351	3.3 ±10.2 ^a^	6.4 (3.6–10.9)	245 (69.8)	85 (24.2)	14 (4.0)	7 (2.0)
4	448	3.8 ± 9.7 ^a^	6.3 (3.0–11.3)	324 (72.3)	94 (31.0)	27 (6.0)	3 (0.7)
5	415	4.3± 9.7 ^a^	6.5 (3.0–11.3)	297 (71.6)	93 (22.4)	20 (4.8)	5 (1.2)
6	472	2.5 ± 9.7 ^ab^	6.0 (2.9–10.7)	346 (73.3)	101 (21.4)	17 (3.6)	8 (1.7)
Total	1938	3.7 ± 9.9 ^a^	6.4 (3.1–11.0)	1386 (71.5%)	435 (22.5%)	89 (4.6%)	28 (1.4%)
Last Exam Before Birth							
1	72	2.6 ± 9.1 ^a^	5.4 (2.5–10.6)	53 (73.6)	18 (25.0)	0	1 (1.4)
2	38	4.5 ± 6.8 ^a^	5.1 (2.9–9.3)	29 (76.3)	9 (23.7)	0	0
3	170	2.5 ± 8.4 ^a^	6.3 (4.1–10.4)	123 (72.4)	44 (25.9)	3 (1.8)	0
4	193	3.0 ± 8.9 ^a^	5.9 (2.9–9.6)	153 (79.3)	31 (16.1)	9 (4.7)	0
5	203	3.5 ± 8.6 ^a^	6.1 (2.8–10.0)	153 (75.4)	42 (20.7)	8 (3.9)	0
6	214	1.8 ± 8.4 ^a^	5.7 (2.5–9.5)	163 (76.2)	47 (22.0)	3 (1.4)	1 (0.5)
Total	890	2.8 ± 8.6 ^a^	5.9 (3.0–9.9)	674 (75.7%)	191 (21.5%)	23 (2.5%)	2 (0.2%)

Total includes 17 additional sonographers with ≤100 exams each. ^a^—significantly different than 0, *t*-test, *p* < 0.05. ^b^—significantly different than Physician 5, Sidak test, *p* = 0.05.

**Table 5 jcm-13-06895-t005:** Cases with recorded fetal sex different than newborn sex.

Case Number	Sonographer Number	Gestational Age at Ultrasound, Weeks + Days	Fetal Sex Reported	Newborn Sex	Review Findings
1	16	36 + 5	Female	Male	Prior exam reported male. No images of genital area on repeat exam; no explanation why sex was changed or why reported.
2.1	8	28 + 5	Male	Female	Images correctly labeled female, error in database and on report.
2.2	8	32 + 2	Male	Female	No images of genital area, sex carried over from prior exam.
2.3	17	36 + 3	Male	Female	No images of genital area, sex carried over from prior exam.
3	13	35 + 1	Female	Male	Images correctly labeled male, error on report.
4	6	32 + 4	Male	Female	Images correctly labeled female, error on report.
5	7	30 + 6	Female	Male	Images labeled “probably female”, recorded in database as female. Patient did not want to know sex, so no sex appeared on report.

## Data Availability

The datasets presented in this article are not readily available because they include protected patient health information and the names of individual healthcare personnel. As an alternative, we provide a sample dataset in Appendix A with anonymized patient and personnel identifiers and random jitter added to dates and measurements. The purpose of providing this data is to facilitate the development and debugging of data analysis programs like those provided in Appendix A. Requests to access the original datasets should be directed to the corresponding author.

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
