# Peer review of "Quantitative Approach to Quality Review of Prenatal Ultrasound Examinations: Estimated Fetal Weight and Fetal Sex"

_jcm, 2024, doi:10.3390/jcm13226895_

Round 1
Reviewer 1 Report
Comments and Suggestions for Authors
This manuscript provides a quantitative approach to quality review of prenatal ultrasound examinations specifically in relation to estimated fetal weight and fetal sex determination.
It is comprehensive and well-written. The future of QA is going to rely on quantitative assessment to identify outliers so that individual reviews with concrete feedback can be specifically directed to individuals.
The suggestions to system manufacturers to incorporate this is excellent because creating a quantitative QA program is outside of the expertise of almost all private practice providers and probably most academic centers unless significant financial and intellectual resources are used to create individual programs.
The part about fetal sex determination is only important as it brings out human error. This same error likely occurs with placental location which may be another area the authors may wish to examine.
Reviewer 2 Report
Comments and Suggestions for Authors
This is a well written, interesting QI project.
It took a couple of reads to identify the clinical purpose of this study: To provide centers with remote referral patients the ability to counsel patients regarding birthweight based on ultrasounds up to 12 weeks before delivery. Emphasizing the importance of how this information can be used clinically would increase the impact/reader interest of this article even if only the conclusion were expanded to provide potential clinical guidance.
Using the same data, the focus of this paper could be changed to focus on sonagrapher/physician education and ultrasound unit standardization.
As it stands this is a solid descriptive article outlining a QI project.
Reviewer 3 Report
Comments and Suggestions for Authors
Thank you for giving me the opportunity of reading the present paper. Although it is a well known and aged topic, the quality of prenatal ultrasound examinations seems to be well conducted in this paper. Furthermore ultrasound identification of fetal growth abnormalities is the mainstay for obstetric management. The present paper adds the identification of pitfalls such as mm/cm in measurements and fetal sex. Suggestions for software developers are well addressed. The identification of operator-specific characteristics in fetal measurements should be transformed in operator-specific curves. Unfortunately longitudinal data are mandatory for a better identification of FGR and LGA. A longer and larger population could add more informations.
